# OpenReview forum: "Steering Large Language Model Activations in Sparse Spaces"
_colmweb.org/COLM/2025/Conference — COLM 2025_

### Official Review · Reviewer_3o12 · 2025-04-30

**Rating:** 4
**Confidence:** 4
**Ethics Flag:** 1

**Summary:**

This paper studies LLM steering by modifying model activations in some way that uses sparse representation learning  to disentangle the dimensions in which to steer. The broad strokes of the idea sound convincing; but the presentation lacks a lot of details and precision and is hard to accurately assess, while the experiments are quite vague, leaving it unclear if/what anything actually works meaningfully better than other approaches.

**Questions To Authors:**

Notation in section 3 and related:
 - what is a "sparse space", as invoked in the abstract and later? I know how to think of sparsity as a property of vectors or matrices, but not of spaces.
 - color coding is introduced on line 113-114 but never used again; all it accomplishes is reducing readability in print.
 -  $\mathbf{a}_{\ell}$ is undefined
 - The shapes of the vector/matrix/tensors in this sections are not defined, or the ranges of values like e.g. "c" in line 1231-132.
 -  Not specified: the form of the sparse encoder $f_\ell$, of the map $\hat{a}_\ell$, and how these functions are obtained,
 -  Sparsity of a vector generally means the property of a vector to have many zeros. Sparse vectors don't need to be non-negative, although non-negativity constraints in optimization do encourage sparsity. However, on line 147 you say "$\sigma$ enforces sparsity constraints (e.g., non-negativity)", which confuses me and needs to be clarified. Are you also enforcing non-negativity in addition to sparsity? This was not said before. And what function is $\sigma$ actually? this is either undefined, or implied notationally to be the sigmoid $1/(1+\exp(-t))$, often denoted sigma, but the latter is quite explicitly never sparse but positive, strictly inside $(0,1)$.
- The heuristics applied on lines 125-131 are not theoretically justified, and not empirically validated. If the sparse activations are allowed to be both positive and negative, then it does not seem correct to zero out indices that are nonzero in the two mean vectors: what if they have opposite directions? If the activations are just positive, then this makes more sense, but why not let the subtraction cancel them out anyway, why manually intervene in the work of the SAE?

On the discussion at the end of 4.1 "Comparing to Dense Activation Steering".
 - For context, this seems to be the only comparison between the proposed method and other methods here.
 - THe curves generally have similar form, maybe up to a different scaling of lambda which could be due to different norms of the corresponding vectors.
 - The claim that steering performance is comparable between the two seems more or less reasonable to me ie the min and max points in the curves are the same.
 - However, the claim that SAS is "more stable and monotonic gains as $\lambda$ increases. In contrast, dense vectors often plateau or even degrade" is not justified by the figures. SAS plateaus and then goes down too from $\lambda=3$ onward.

Other points
 - The color of lines in fig 5 and 6 are hard to distinguish, maybe vary line style or markers too, and reduce the whitespace around the figures.
 - In section 4.5, it seems just natural to me that encoding the same data with a wider sparse autoencoder will lead to sparse features. Is this in any way specific to the LLM activation data or just an inherent outcome of the SAE algorithm used? I assume the latter, and of course this can be controlled through various regularization techniques, i.e., a parameter of the SAE algorithm.
- The fact that benchmark performance doesn't seem impacted by autoencoder width also doesn't seem meaningful, maybe a more important metric here to check would be whether the autoencoders have constant / close to perfect reconstruction error at these widths too. In general I don't see what this section says beyond the obvious, and I don't see any connection between it and "scaling laws" as invoked.

**Reasons To Accept:**

The broad strokes of the idea to learn sparse representations of activations makes some intuitive sense and the method does seem to work to some extent.

**Reasons To Reject:**

- There seems to be no meaningful comparison with the dense case or other alternative approaches beyond Figure 4, which at best shows the method is comparable to the dense case empirically.
- A lot of mathematical symbols are undefined and often unclear, and the method section is this hard to concretely understand.
- Evaluation in 4.2 is only through "LLM-as-a-judge", without any additional human validation or any other metrics to cross-validate against. In contrast, evaluation in 4.3 is just in terms of correct answer accuracy, without any quality assessment of the generated text. Overall this to me makes the evaluation too one-dimensional: in controlled generation, usually there is some trade-off between some sort of quality/fluency and satisfaction of the desired controlled attribute, but we cannot see this when measuring a single attribute.
- 4.4 seems artificial and moreover lacks comparison to any other approaches; while 4.5 is mostly speculatory with any reported findings obvious.

---

> ### Author Response · Authors · 2025-06-03
> **Response 1 of 2**
>
> We appreciate your thorough review and the constructive feedback provided. Below, we address each of your concerns in detail.
>
> Before diving into individual points, we would like to clarify a few aspects of our work that may not have been sufficiently emphasized in the current draft:
>
> - **SAS effectiveness**: Our goal is not to outperform dense steering but to match its effectiveness while gaining interpretability and control.
> - **Sparse space and notation**: “Sparse space” is a standard term and been used before [1]; here it refers to high-dimensional vectors in which most dimensions are zero. In the SAE literature, sparsity is constrained using non-linearities such as ReLU (or in our case JumpReLU), which enforce both *sparsity* and *positivity*. We clarify that $\sigma$ refers to JumpReLU, not sigmoid.
> - **SAE details**: We use pretrained SAEs from Gemma Scope [2]; we do not train them ourselves.
> - **Evaluation diversity**: We included multiple-choice, open-ended, and effectiveness evaluations on standard benchmarks (MMLU and TruthfulQA). The “LLM-as-a-judge” approach is now common in recent work (e.g., Zheng et al., 2023, *Judging LLM-as-a-Judge with MT-Bench and Chatbot Arena*).
> - **Detailed algorithmic steps**: While Figures 1 and 2 should help visualize the method, detailed algorithmic steps are provided in *Appendices E and F*.
>
>
>
> ### 1. Comparison with Dense Activation Steering
>
> In Section 4.1, we present a comparative analysis between SAS and dense steering, focusing on behavioral modulation. Figure 4 shows SAS achieves comparable performance. Our goal is not stronger steering, but more *interpretable* and *controllable* behavior modulation, so matching dense performance is meaningful.
>
>
> ### 2. Clarity of Notations
>
> We will revise the paper to:
>
> - Define all mathematical symbols and tensor dimensions.
> - Clarify that $f_l$ and $a_\ell$ refer to encoder/decoder functions from **pretrained** SAEs.
> - Explicitly state that $\sigma$ is JumpReLU.
> - Clarify "sparse space" as the latent space of the SAE with many zeros, as commonly used in sparse coding literature.
>
>
>
> ### 3. Evaluation Metrics
>
> While we primarily use LLM-as-a-judge, this has become a standard scalable metric.
> From Zheng et al., 2023, "Judging LLM-as-a-Judge with MT-Bench and Chatbot Arena":
> > “LLM-as-a-judge is becoming standard practice in LLM evaluation pipelines due to cost and scalability.”
>
> From Chiang et al., 2023, "Can LLMs be Better Judges than Humans":
> > ”We find that LLMs often provide more consistent and informative feedback than humans.”
>
> However, we agree that additional human evaluations or alternative metrics would be valuable, and we plan to explore these in future work. Nonetheless, a thorough set of generation examples for each steering behavior is provided in *Appendix N, “Open-Ended Generations Examples.”*
>
>
> ### 4. Sections 4.4 and 4.5 Evaluations
>
> We view these sections as important for understanding not just whether SAS works, but also why it works and how it can be composed and scaled. They show:
>
> - Different behaviors can be **composed** using SAS vectors to have simultaneous effect.
> - Wider SAEs show a trend **toward monosemantic** SAS vectors. While the number of raw extracted features remains roughly the same—especially beyond a width of 65K (Figure 7, middle)—only a few correspond to SAS (Figure 7, left), indicating that concepts are becoming increasingly monosemantic.
>
>
> ### 5. Specific Questions and Clarifications
>
> - **"Sparse space"**: Latent space where SAE outputs have mostly zero entries.
> - The same **color coding** is used in all figures, especially in Figures 1 and 2.
> - **Symbols and shapes**: Will be explicitly defined in the revision.
> - **Encoder/decoder forms**: From pretrained Gemma Scope SAEs (JumpReLU-based, for Gemma-2B/9B).
> - **Sparsity vs. non-negativity**: Non-negativity is enforced by JumpReLU in the pretrained SAE; we use the same activation function since the decoder expects non-negative codes.
> - **Feature filtering heuristics**:
>     -  **Threshold $\tau$**: Vary the number of active features (i.e. controlling sparsity); ablation provided in *Appendix C.2*.
>     -  **Removing common features**: Because common features almost cancel each other out and are concentrated around zero, removing them further makes the final SAS vector sparser with no performance downgrade; ablation provided in *Appendix C.4*.
>     -  **Subtracting the two final vectors**: This further helps reinforce the desired behavior and suppress the model tendencies already present in the activation; ablation provided in *Appendix C.3*.

---

> > ### Author Response · Authors · 2025-06-03
> > **Response 2 of 2**
> >
> > ### 6. Figure 4 Analysis
> >
> > Our intent was to show SAS maintains performance over a range of scaling values. We agree that calling it “monotonic” may overstate the trend and will revise the phrasing accordingly.
> >
> >
> > ### 7. Additional Points
> >
> > - **Figure readability**: We will improve line styles and layout in Figures 5 and 6.
> > - **Section 4.5**: While sparsity increases with SAE width, our key result is that interpretability (monosemanticity) of SAS vectors improves.
> > - **A better metric**: Figure 4 in Gemma Scope \[1] shows that wider SAEs achieve lower reconstruction error.
> >
> > ---
> >
> > Thank you again for your detailed and thoughtful review. We are happy to clarify further and look forward to continuing the conversation.
> >
> >
> > ---
> > [1] Fard, Mahdi Milani, et al. "Compressed least-squares regression on sparse spaces." Proceedings of the AAAI Conference on Artificial Intelligence. Vol. 26. No. 1. 2012.
> >
> > [2] Lieberum, Tom, et al. "Gemma scope: Open sparse autoencoders everywhere all at once on gemma 2." arXiv preprint arXiv:2408.05147 (2024).

---

> > > ### Comment · Reviewer_3o12 · 2025-06-06
> > > **Thank you for your answers**
> > >
> > > I am happy you agree the room for increasing clarification. While I would prefer to see another revision due to the big gap in changes needed, I will increase my score slightly. I cannot assess the paper higher than this without more substantial revisions however.
> > >
> > > Regarding LLM-as-judge, the Zheng paper does seem to require quite a lot of maneuvering to avoid LLM biases. The other reference you list "Chiang et al., 2023, "Can LLMs be Better Judges than Humans"" -- I could not find this on the internet at all. Instead, I found other work that shows more negative and nuanced findings: https://arxiv.org/abs/2406.18403v3.
> > >
> > > I still believe the paper is below the bar due to the very limited amount of empirical comparisons/ablations (I am not convinced that simpler methods don't also have the benefits of yours) and limited novelty overall. The experimental design of this work is in my opinion insufficient for drawing scientific conclusions about activation steering. I am convinced by the approach, but I don't think the experimental setup justifies the claims.

---

> > > > ### Author Response · Authors · 2025-06-06
> > > >
> > > > Thank you for engaging in the discussion and for raising your score. With respect, we see things differently on certain points, particularly regarding our empirical evaluation and the novelty of our contribution.
> > > >
> > > > ### On the Use of LLM-as-a-Judge
> > > >
> > > > We agree that LLM-based evaluation has limitations. However, it is a **standard methodology** in lack of other metrics that are not trivial how to come up with. We mentioned *[Zheng et al., 2023](https://arxiv.org/abs/2306.05685)* as a benchmark reference with more than 3900 citations, but LLM-as-a-judge is also used in contemporary works like [Yuan, W., et al., 2024]( https://arxiv.org/abs/2401.10020). We also thank you for highlighting *[Bavaresco et al., 2024](https://arxiv.org/abs/2406.18403v3)*. We agree with its conclusions on evaluation bias and note that our findings are consistent with its recommendation to report metrics under multiple setups, which we do in Section 4.1 (Multi-Choice Questions Steering Evaluation), Section 4.2 (Open-Ended Generation Steering Evaluation), and Section 4.3 (Steering Effect on Standard Benchmarks).
> > > >
> > > > Sorry for the slight error in the title of the paper. The paper we are referring to is [Chiang et al., 2023, "Can Large Language Models Be an Alternative to Human Evaluations?"](https://arxiv.org/abs/2305.01937)
> > > >
> > > > ---
> > > >
> > > > ### On Experimental Rigor
> > > >
> > > > Your claim that "simpler methods might enjoy the same benefits" is speculative. In contrast, our work provides **explicit ablations and analyses** that support SAS as a scalable, interpretable alternative to dense steering:
> > > >
> > > > - **Figure 4**: Direct comparison with dense steering shows **matching performance**.
> > > > - **Table 1**: SAS supports **behavioral compositionality**. In a dense setting, we cannot think of any ways to compose the steering vectors with new features as the dense representations do not have any interpretation.
> > > > - **Figure 7**: Scaling SAE width improves monosemanticity (qualitatively and quantitatively), a key hypothesis for interpretability and feature disentanglement.
> > > >
> > > > ---
> > > >
> > > > ### On Novelty of Sparse Activation Steering (SAS)
> > > >
> > > > Our method is not a trivial variant of prior work. Previous works by [Mayne, H., et al., 2024](https://arxiv.org/abs/2411.08790) has shown how challenging this is, and naive attempts to sparsify dense steering vectors fail. SAS is a novel approach to combine contrastive prompt pairing with sparse autoencoders to produce interpretable steering vectors. It enables direct interpretability of features, compositional control, and a fully **plug-and-play mechanism** for activation editing without fine-tuning.
> > > >
> > > > ---
> > > >
> > > > ### Final Note
> > > >
> > > > We respectfully believe that the paper already addresses these concerns: the experiments throughout, along with feedback from the other reviewers, support our claims. That said, we would be grateful for the opportunity to discuss any remaining questions and clarify anything that might be standing in the way of a higher score.

---

### Official Review · Reviewer_9Lrh · 2025-05-12

**Rating:** 8
**Confidence:** 4
**Ethics Flag:** 1

**Summary:**

This paper introduces Sparse Activation Steering (SAS), a novel method for guiding the behavior of large language models (LLMs) in sparse latent spaces.
The motivation stems from the challenges in controlling LLMs and the limitations of existing methods such as instruction fine-tuning, prompt engineering, and reinforcement learning from human feedback. Activation steering, which modifies internal model activations during inference, has potential but faces issues in dense activation spaces due to superposition. Sparse representations offer a solution, and SAS aims to leverage them.
The method involves generating SAS vectors through a contrastive prompt-pairing approach. It creates prompt pairs for a target behavior, extracts sparse representations using a sparse autoencoder (SAE), filters out inactive features, removes shared features, and computes the final steering vector. During inference, the SAS vector is added to the sparse latent representation of the generated token to steer the model's output.
The experiments cover multiple-choice questions, open-ended generation tasks, and standard benchmarks. The results show that SAS is as effective as its dense-space counterpart in steering and has advantages in interpretability and compositionality. It can also enhance model performance on certain benchmarks. Additionally, the study on feature compositionality demonstrates the ability to combine different behaviors, and the scaling study shows that increasing the SAE's dictionary size improves the sparsity of SAS vectors, approaching ideal monosemanticity.

**Reasons To Accept:**

The paper is well-structured and presents a comprehensive study. The work is solid, with detailed explanations of the method, experiments, and results. It provides a valuable contribution to the field of LLM behavior control.

1. The concept of using sparse spaces for activation steering is clearly presented. The contrastive prompt-pairing approach for generating SAS vectors is intuitive and effectively separates behavior-specific features.
2. The experiments are well-designed and cover a wide range of tasks, including multiple-choice questions, open-ended generation, and standard benchmarks. The use of pre-trained SAEs in the Gemma-2  models simplifies the implementation and allows for focused exploration of the proposed method.
3. The demonstrated interpretability and compositionality of SAS vectors are significant advantages. The ability to isolate and combine features in sparse spaces provides more fine-grained control over LLM behavior, which is a notable improvement over existing methods.

**Reasons To Reject:**

1. While the combination of activation steering and sparse autoencoders is clever, there may be concerns about the level of innovation. The method seems to build on existing techniques in both areas. However, the way it addresses the superposition problem in dense spaces by leveraging sparse representations is novel. It offers a new perspective on LLM behavior control, and the empirical results show its effectiveness, which somewhat mitigates the innovation concern.
2. The experiments could be extended to larger-scale models. Although the authors mention potential limitations due to existing SAEs and research resources, larger-scale experiments would provide more conclusive evidence of the method's performance. For example, testing on more diverse and larger-scale LLMs could show the method's generalizability better. However, the current results still offer valuable insights, and the limitations are understandable in the context of the research.

---

> ### Author Response · Authors · 2025-06-03
>
> Thank you very much for your thoughtful and encouraging review. Your remarks on the interpretability and compositionality advantages of Sparse Activation Steering (SAS) are especially motivating for us.
>
> **Response to your main suggestion — larger-scale models**
>
> **New experiments now available.** As noted in your comments, additional evidence on larger and more diverse LLMs would strengthen the work. Therefore, we have experimented with the multiple-choice steering experiments on the newly released Llama-3.1-8B SAE checkpoints from the Llama-Scope project [1]. The results closely track those reported for Gemma models. A preliminary figure is included below; the full experimental details (hyperparameters, and qualitative outputs) will appear in our next revision.
>
> **New Llama results:** \[[reference link](https://ibb.co/x8DKGLjT)]
>
> ---
>
> [1] He, Zhengfu et al. “Llama Scope: Extracting Millions of Features from Llama-3.1-8B with Sparse Autoencoders,” 2024.

---

### Official Review · Reviewer_QAAq · 2025-05-13

**Rating:** 7
**Confidence:** 3
**Ethics Flag:** 1

**Summary:**

This paper introduces Sparse Activation Steering (SAS)  controlling the behavior of LLMs at inference time by directly modifying sparse internal activations. Traditional activation steering techniques operate in dense spaces, which suffer from superposition, making it difficult to achieve fine-grained and interpretable control. SAS addresses this by leveraging Sparse Autoencoders to project activations into a sparse, disentangled latent space, enabling more interpretable.  The methodology is sound and well-motivated. The paper is clearly written and many implementation details are clearly provided.

**Questions To Authors:**

Although the SAS is inference time steering method, but the steering vector still training. How is the computational cost? Is there any trade-off between steering scope exploration and efficiency? Can the authors comment on latency, memory, and feasibility for real-world deployment, since SAE dictionary sizes scale to hundreds of thousands or even millions, what are the computational implications during inference?

**Reasons To Accept:**

1. Unlike prior dense steering methods, this paper introduces the Sparse Activation Steering for influencing LLM behavior using sparse latent representations.
2. Extensive experiments demonstrate that SAS performs on par with dense steering methods in both multiple-choice and open-ended tasks. It also shows consistent gains on standard benchmarks like TruthfulQA and MMLU under moderate steering.
3. The paper is clearly written, provides detailed descriptions and ablation studies.

**Reasons To Reject:**

1.  The experiment part lacks generalisability evaluations. The evaluation primarily focus on specific alignment tasks with a limited backbone (Gemma).
2. While the paper demonstrates strong empirical results, it lacks deeper theoretical analysis regarding why steering in sparse spaces yields improved modularity and compositionality.

---

> ### Author Response · Authors · 2025-06-03
>
> Thank you for the thoughtful review. Below, we address your concerns and questions.
>
> ---
>
> > Limited backbone (Gemma).
>
> **New experiments now available.** Recently, the **Llama-Scope** project \[1] released SAE checkpoints for the Llama-3.1-8B model. We have reproduced the multiple-choice SAS-steering experiments on these checkpoints; the trends mirror those in Gemma (see preliminary plot below ($\tau=0.2, \lambda=\pm1$)). We will ablate more on these models in the next revision.
>
> **New Llama results:** \[[reference link](https://ibb.co/x8DKGLjT)]
>
>
> > Why steering in sparse spaces yields improved modularity and compositionality.
>
> From *Toy Models of Superposition* \[2] we learn that when a network is forced to represent many sparse input features in too few dimensions, it stores them in **superposition**—several concepts share a single neuron, so an intervention on one concept inadvertently affects others. Once a sparse autoencoder (SAE) wrapped around, it discovers a sparse basis that separates these features, those different concepts become linearly independent directions.
>
> Sparsity is therefore the mechanism that resolves superposition. Resolving superposition yields a clean dictionary of features, and a clean dictionary makes feature edits modular and their compositions linear—hence the improved modularity and compositionality we observe when steering in sparse spaces.
>
> ---
>
> ### Questions
>
>
> > Cost of *building* a steering vector
>
> To clarify, *no gradient‐based training* is performed when we build a Sparse Activation Steering (SAS) vector. For each behaviour, we simply run the model once on a small contrastive dataset (≤ 1,000 prompt pairs) and obtain the sparse representation with the SAE encoder.
>
> | Phase                   | What runs?                                     | Wall-time on Gemma-2 2B (A100-40 GB) | Comment                                                             |
> | ----------------------- | ---------------------------------------------- | ------------------------------------ | ------------------------------------------------------------------- |
> | **Vector construction** | Forward pass on ≤ 1,000 prompts + SAE *encode* | < 4 min / behaviour                  | Same cost class as dense CAA, which also touches each example once. |
>
> > Cost at *inference*
>
> At inference, we load **one** SAE (encoder + decoder) for a chosen layer and perform:
>
> 1. **SAE encode → K** active features
> 3. Add $\lambda \times v_{b, \ell}$ to the sparse representation
> 4. **SAE decode** (plus the $\Delta$ correction)
>
> This adds a single extra general matrix–matrix multiplication at one layer:
>
> | SAE width     | Extra parameters          | Extra VRAM (FP16) |
> | ------------- | ------------------------- | ----------------- |
> | 65 K          | 2 × 2304 × 65,000 ≈ 300 M | \~ 0.6 GB         |
> | 262 K         | 1.1 B                     | \~ 2.2 GB         |
>
> Gemma-2 2B's residual stream dimension is 2304.
>
> > Practical comments (Exploration and Efficiency)
>
> * Exploring different layers or $\tau$-thresholds only requires re-scoring already-cached sparse activations—no extra passes through the model.
> * Wider SAEs yield sparser, more monosemantic vectors without hurting steering power, so practitioners can trade a modest RAM increase for better interpretability.
> * If a tighter budget is needed, the SAE can be 8-bit-quantized.
>
> ---
>
> [1] He, Zhengfu et al. “Llama Scope: Extracting Millions of Features from Llama-3.1-8B with Sparse Autoencoders,” 2024.
>
> [2] Elhage, Nelson, et al. "Toy models of superposition." arXiv preprint arXiv:2209.10652 (2022).

---

> > ### Comment · Reviewer_QAAq · 2025-06-09
> > **Official Comments by Reviewer QAAq**
> >
> > Thanks for your response.
> > Most of my question are resolved from your response. I prefer to keep my original rating.

---

> > > ### Author Response · Authors · 2025-06-09
> > >
> > > Thank you again for your time and insightful feedback. We will ensure that your constructive comments and questions, as well as the newly added experiments on the Llama model, are reflected in the next revision of the paper.

---

### Official Review · Reviewer_siLA · 2025-05-20

**Rating:** 6
**Confidence:** 4
**Ethics Flag:** 1

**Summary:**

The paper introduces Sparse Activation Steering (SAS), a technique for editing activations in sparse latent spaces to either elicit or suppress specific downstream behaviors. SAS involves a contrastive prompt-pairing approach for extracting corresponding activations which are then passed through a SAE to obtain a steering vector via difference-in-means, that can be applied to new inputs for controlling the target behavior. The study evaluates SAS on both Multiple Choice as well as Open-Ended Generation settings across seven key behaviors. It also examines to what degree features that are found to correspond to distinct behaviors or concepts can be combined to exert compositional effects on downstream generation. Finally it looks into how monosemanticity scales with SAE width, while also conducting a preliminary evaluation of SAS on transcoders.

**Questions To Authors:**

- From which token(s) of the input prompts are the activations extracted from to generate the SAS vectors?
- How do you explain that steering for Sycophancy improves performance in MMLU (Figure 6)? (I couldn't find any Sycophancy example prompts)
- In lines 231-232 you mention "steering toward Refusal and away from Hallucination consistently improves factual correctness". I assume this is meant by means of each behavior separately as shown in Figure 6 (right). Did you try to assess this in the context of feature compositionality ie weighting the Hallucination and Refusal steering vectors accordingly?
- What are some advantages of SAS compared to SPARE(https://arxiv.org/abs/2410.15999)?
- For the preliminary evaluation on transcoders in Appendix D you are using t=0.2 for generating the SAS vectors whereas for the SAE
  based vectors you are using higher values of t=0.7, what is the reason for this?
- [Suggestion] It might be useful to have corresponding colors for each behavior in Figure 3 (as in Figure 5) because it is not immediately obvious which curve is for which behavior.

**Reasons To Accept:**

- The paper is very well written, well presented and easy to follow
- It showcases the effect of SAS on standard benchmarks (MMLU, TruthfulQA),
  covering a wide range of key behaviors (refusal, sycophancy, hallucination, corrigibility,
  AI Coordination, survival instinct and myopic reward)
- It studies feature compositionality providing interesting insights to the community

**Reasons To Reject:**

- SAS is evaluated on a limited range of models - primarily on Gemma models.
- SAS is compared to a limited range of representation engineering techniques (only CAA)
- The section on feature compositionality is interesting but it is rather limited in scope. It is built around a particular setting:
  the interaction between the complex behavior of Myopic Reward (SAS vector) and the more specific concept of gender bias (non SAS
  steering vector) for which it showcases modular control over the individual behaviors. This being the only setting that is examined raises
  questions to the wider applicability of this sort of modular control to other cases.

---

> ### Author Response · Authors · 2025-06-03
>
> Thank you for the thoughtful review. Below, we address your concerns and questions.
>
> ---
>
> > Limited range of models.
>
> * **What is already in the submission.** *Appendix D (Transcoder)* reports multiple-choice results on **Llama-3.2 1B** using **Transcoder** sparsifiers (an alternative to SAEs that operate only on the MLP block). This preliminary result shows the same trend as in Gemma models; however, MLP representations are less expressive than the residual stream, and broader experiments are needed to draw firmer conclusions.
> * **New experiments now available.** Recently, the **Llama-Scope** project \[1] released SAE checkpoints for the Llama-3.1-8B model. We have reproduced the multiple-choice SAS-steering experiments on these checkpoints; the trends mirror those in Gemma (see preliminary plot below ($\tau=0.2, \lambda=\pm1$)). We will ablate more on these models in the next revision.
>
> **New Llama results:** \[[reference link](https://ibb.co/x8DKGLjT)]
>
> ---
>
> > Limited range of representation-engineering techniques.
>
> Our goal is to show that *contrastive steering*—originally implemented in dense space by CAA—transfers to *sparse* space **without losing efficacy** while *gaining* interpretability (sparsity, feature compositionality, and progress toward monosemanticity).
>
> ---
>
> > Limited scope of compositionality.
>
> We agree that there is a lack of a systematic **compositionality benchmark**. While our experiment shows the effectiveness of SAE-steering on this particular task, creating such a benchmark is a natural next step, and we will highlight this explicitly in the future work section.
>
>
> > Why the combination of SAS and non-SAS vectors?
>
> To clarify our compositionality study: it is designed to probe one specific question—*Can a complex behaviour (e.g., **Myopic Reward**) be cleanly combined with a simple, atomic feature (e.g., **Gender**) ?*
>
> This choice follows directly from the motivation in the introduction:
>
> * Prior work that labels individual SAE features assumes **monosemanticity** (one meaning per dimension), such as the non-SAS steering vectors from Neuronpedia \[2].
> * In practice, (i) many dimensions remain unlabeled, and (ii) complex behaviours such as *myopic reward* rarely align with a single feature. They are typically represented by a **small bundle** of partially disentangled features because the SAE has not yet fully resolved superposition—especially at smaller scales, as discussed in Section 4.5 (*Scaling Monosemanticity*).
> * Conversely, simple and highly frequent attributes like **gender** *can* be captured by a single, confidently labeled feature.
>
> Therefore, the Myopic × Gender case illustrates the **mixed toolkit** we expect practitioners to use in the sparse setting:
> - derive SAS vectors for complex behaviours, and
> - reuse well-labeled non-SAS (i.e., from Neuronpedia) features for atomic attributes.
>
> Finally, note that the same SAS approach could also identify gender-specific features, but doing so would require constructing a small contrastive dataset.
>
> ---
>
> ### Questions
>
> > Which token representation.
>
> We always take the *last token* of the prompt—the hidden state used to predict the first generated token.
>
> > Sycophancy and MMLU (Figure 6).
>
> Sycophancy steering nudges the model toward answers that closely match the majority-held, human-endorsed view of each question. While deeper investigation is required, one hypothesis is that, because MMLU’s ground-truth labels correspond to these majority-endorsed answers, sycophancy increases the probability of selecting them.
>
> > Compositionality of Hallucination and Refusal.
>
> Although we did not show an explicit *linear-combination* experiment in the main text, we analyzed the relationship between these two behaviours and every other pair in **Appendix L (Behavioural Correlation), Figure 25d**.
> That figure measures, for every pair of SAS vectors, the fraction of features they share. *Refusal(+)* and *Hallucination(–)* exhibit one of the highest overlaps, which explains why steering with either vector alone already improves factual correctness in Figure 6 (right).
>
> > AS compared to SPARE?
>
> SPARE uses SAEs to resolve *knowledge conflicts*—that is, it decides which knowledge source the model should trust when the current context contradicts what is stored in memory. SAS, on the other hand, is designed to steer an LLM’s *behavioural traits* at inference time. Because the two methods address different goals, they are complementary rather than directly comparable.
>
> > Why using $\tau = 0.2$ for transcoders.
>
> Pre-trained transcoders are far sparser (\~32 active features) than SAEs (\~150). A lower $\tau$ therefore preserves a comparable *absolute* number of features. We will add this rationale in Appendix D.
>
>
> ---
>
> [1] He, Zhengfu et al. “Llama Scope: Extracting Millions of Features from Llama-3.1-8B with Sparse Autoencoders,” 2024.
>
> [2] Johnny Lin. *Neuronpedia: Interactive Reference and Tooling for Analyzing Neural Networks*, 2023.

---

> > ### Author Response · Authors · 2025-06-09
> >
> > Dear Reviewer,
> >
> > Thank you again for taking the time to provide valuable feedback on our paper. As the discussion period concludes tomorrow, we kindly ask if you’ve had a chance to review our responses.
> >
> > We hope we have adequately addressed your concerns and clarified any ambiguities. Should you have any further questions or suggestions, we would be happy to respond promptly.
> >
> > Additionally, if our responses and the newly added experiment have satisfactorily resolved your concerns, we would greatly appreciate it if you could reflect this in your review score.
> >
> > Thank you once again for your time and consideration.

---

> > ### Comment · Reviewer_siLA · 2025-06-09
> >
> > Thank you for your clarifications. I believe that your work is a positive contribution to the community.
> >
> > However, I respectfully disagree with your position that SPARE ( Sparse Auto-Encoder-based Representation Engineering ) [1] is not directly comparable to your method as both methods fall within the representation engineering domain, regardless of the dowstream task they are applied to. I'm not sure why knowledge selection should not be considered a behavioral trait of LLMs that can be steered during inference. Also I want to clarify that SPARE does not decide which knowledge source the model should trust, rather it offers the means to steer towards one or the other source. From the "Limitations" section of [1]:
> >
> > "Finally, the control over knowledge selection behaviours is evaluated primarily in terms of steering the model towards either contextual or parametric knowledge. In practice, the decision about which knowledge source to trust may not be binary or require a more elaborate approach, such as using another model as a critic (Hong et al., 2024)"
> >
> > Given that more extensive comparisons would help better showcase the particular advantages of SAS compared to other methods as well as the limited scope of the (very insteresting) compositionality benchmark, I intend to keep my score.
> >
> > ---
> > [1] Yu Zhao, Alessio Devoto, Giwon Hong, Xiaotang Du, Aryo Pradipta Gema, Hongru Wang,
> > Xuanli He, Kam-Fai Wong, and Pasquale Minervini. Steering knowledge selection be-
> > haviours in llms via sae-based representation engineering (https://arxiv.org/abs/2410.15999)

---

> > > ### Author Response · Authors · 2025-06-09
> > >
> > > Thank you for your constructive feedback and for considering our work a positive contribution. We appreciate your engagement in the discussion.
> > >
> > > Regarding SPARE, we agree that it is an interesting work that falls under the domain of representation engineering but is designed for different downstream tasks than SAS. To clarify, SPARE identifies discrepancies between a model's stored knowledge and contextual information, focusing on changing the model's preference between in-context and stored knowledge at test time, while SAS provides a general framework for steering model output toward any desired behaviors.
> > >
> > > We're glad you found our compositionality analysis interesting and are actively considering the development of a comprehensive benchmark on this topic in future work.
> > >
> > > In the next revision, we will:
> > > - Discuss the new experiments on the LLama model
> > > - Clarify the distinction between SPARE and SAS
> > > - Reflect changes based on your detailed questions to address potential concerns from other readers as well

---

### Decision · Program_Chairs · 2025-07-08

**Decision:**

Accept

**Comment:**

The paper seems to be well-received overall, even if it has some weaknesses. The authors justified their decisions (such as not comparing against a baseline like SPARE). The topic seems to be an overall "hot topic" at the moment, and I think the paper could complement the program at the conference well.